# Is High-Intensity Interval Training Suitable to Promote Neuroplasticity and Cognitive Functions after Stroke?

**DOI:** 10.3390/ijms22063003

**Published:** 2021-03-16

**Authors:** Nicolas Hugues, Christophe Pellegrino, Claudio Rivera, Eric Berton, Caroline Pin-Barre, Jérôme Laurin

**Affiliations:** 1INMED, INSERM, Aix-Marseille University, 13007 Marseille, France; nicolas.HUGUES@univ-amu.fr (N.H.); christophe.pellegrino@inserm.fr (C.P.); claudio.rivera@inserm.fr (C.R.); 2CNRS, ISM, Aix-Marseille University, 13007 Marseille, France; eric.berton@univ-amu.fr (E.B.); caroline.PIN-BARRE@univ-amu.fr (C.P.-B.)

**Keywords:** stroke rehabilitation, cerebral ischemia, cognition, endurance exercise, neurotrophic factors, neurogenesis, angiogenesis, rat and human model

## Abstract

Stroke-induced cognitive impairments affect the long-term quality of life. High-intensity interval training (HIIT) is now considered a promising strategy to enhance cognitive functions. This review is designed to examine the role of HIIT in promoting neuroplasticity processes and/or cognitive functions after stroke. The various methodological limitations related to the clinical relevance of studies on the exercise recommendations in individuals with stroke are first discussed. Then, the relevance of HIIT in improving neurotrophic factors expression, neurogenesis and synaptic plasticity is debated in both stroke and healthy individuals (humans and rodents). Moreover, HIIT may have a preventive role on stroke severity, as found in rodents. The potential role of HIIT in stroke rehabilitation is reinforced by findings showing its powerful neurogenic effect that might potentiate cognitive benefits induced by cognitive tasks. In addition, the clinical role of neuroplasticity observed in each hemisphere needs to be clarified by coupling more frequently to cellular/molecular measurements and behavioral testing.

## 1. Introduction

Stroke is one of the non-communicable diseases with the greatest number of disability-adjusted life years reflecting health loss [1]. Cognitive impairments, including attention, memory, executive functioning and information processing deficits, frequently contribute to reduce the quality of life, notably by doubling the risk of developing dementia [2,3,4]. A decline in cognitive skills is also strongly predictive of the inability to return to work, thereby contributing to the socioeconomic burden of stroke [5]. Currently, stroke rehabilitation remains crucial to counteract cognitive impairments.

Numerous cognitive training strategies are employed in the clinic such as the use of diary, prompting devices, computers as well as remedial strategies through virtual reality, gaming and several memory tasks [6,7]. Some of these strategies can reduce attention deficits just as verbal, prospective and working memory impairments [8]. However, the observed improvements in laboratory experiments could not systematically be transferable to daily life cognitive tasks [9]. Another limitation of training strategies is related to patients inability to preserve long-term cognitive benefits [10]. Moreover, some studies failed to find any positive effects on both memory and executive functions [8,9]. Therefore, the cognitive rehabilitation guidelines need to be reconsidered after stroke [8].

Beyond its roles in cardiorespiratory and muscular functions, endurance training can also be considered a part of cognitive rehabilitation in individuals with stroke [8]. Moderate-intensity continuous training (MICT), the most investigated exercise regimen, could increase attention, information processing speed and implicit memory performance in patients with stroke [11,12,13]. In rodents with cerebral ischemia, endurance training might also improve cognitive functions by stimulating synaptic plasticity, neurogenesis and angiogenesis through the upregulation of neurotrophins levels [[8],[11],[14][15],[16],[17],[18]]. However, cognitive improvements are not systematically observed with MICT in both stroke patients and rodents with cerebral ischemia (also in healthy individuals), although it frequently improves aerobic capacity [12,13,19,20,21,22,23].

Exercise regimens with higher intensities, such as high-intensity interval training (HIIT), emerge as encouraging alternatives to improve cardiovascular and brain health following stroke [24,25,26,27]. HIIT, the most popular trend of 2018 [28], involves repeated short-to-long bouts of high-intensity exercise interspersed with active or passive recovery periods [29]. HIIT is defined as short (<1 min) or long (1–5 min) series performed above the lactate/ventilatory threshold suggesting an accumulation of lactate during sessions contrary to MICT [29,30,31]. HIIT is feasible in individuals with stroke without any signs of cardiovascular intolerance or arrhythmias [26,32,33]. Moreover, HIIT is considered to be enjoyable and a time-efficient strategy to improve wellbeing and cardiovascular and muscular functions [28,32,33,34,35,36]. In individuals with stroke, recent evidence indicates a potential role of HIIT by measuring circulating molecular markers of neuroplasticity that might improve cognition functions [24,25,37,38]. In rodents with cerebral ischemia, HIIT, which induced strong improvements in aerobic parameters, also upregulated neuroplasticity markers in the hippocampus and the cortex when initiated during the therapeutic window (the 2 first weeks poststroke) [37,38,39]. However, the link between neuroplasticity and cognitive outcomes following HIIT remains to be defined in both preclinical and clinical studies, thereby explaining why exercise guidelines for brain health remain inexistent in individuals with stroke.

The purpose of this review is to examine whether HIIT could be suitable for promoting neuroplasticity processes and/or cognitive functions after stroke by discussing findings from molecular to behavioral levels in both human and animal studies. The various methodological limitations in exercise studies, which could explain controversial findings, need to be first considered in this review to clarify both the clinical relevance of rodent studies and the impact of clinical studies on the exercise recommendations. Then, this review highlights the relevance of HIIT in promoting neuroplasticity and/or cognitive functions from single bout of HIIT protocols to training HIIT protocols in both healthy populations (young and older) and individuals with stroke. In addition, the preventive role of HIIT on the stroke severity in rodents (by starting HIIT program before stroke onset) is also discussed. To reinforce the potential role of HIIT in stroke rehabilitation, it is important to discuss the relevance of combining HIIT with cognitive tasks to potentiate their effects on neuroplasticity and cognitive performances.

## 2. Methodological Considerations for Endurance Exercise Studies

In both rodent and human studies, several methodological limitations keep us from finding optimal endurance programs to recover both cognitive and sensorimotor functions in individuals with stroke. These limitations are related to the heterogeneity of studied populations, the small number of patients, the variability of exercise types (over-ground, treadmill and cycling in humans and treadmill, running wheel and swimming in rodents) [40], the timing of rehabilitation [41] and the lack of determination of accurate aerobic exercise parameters, also named the FITT principle, i.e., frequency, intensity, time and type [42]. Additionally, a recent meta-analysis indicated that very few clinical studies have investigated more than one type of exercise [21] limiting our ability to ensure suitable exercise doses for cognitive benefits.

### 2.1. Definition of Exercise Intensity

The intensity of repeated bouts of HIIT is above the lactate threshold (or at 85–90% of maximal speed or HR_max_) separated by active (i.e., 30–50% of maximal speed or HR_max_) or passive recovery periods [26,29,43]. However, the exercise intensity can strongly differ from these recommendations/guidelines in previous studies [14,32,40,44,45]. Indeed, high intensity is referred to maximum-tolerated treadmill speed in some clinical studies [33]. However, the physiological status of subjects (blood lactate concentration, % of VO_2max_, % of maximal heart rate or HR_max_) during training sessions is not mentioned, despite the fact that it indicates the intensity level reached by patients. In other studies, “vigorous exercise” is defined as an exercise intensity sufficient to produce sweat [46,47], whereas sweating depends on many factors such as exercise duration, environmental temperature, psychological state, genetic factors and fitness levels.

In preclinical rodent studies using swimming, maximal effort is related to the intensity at which rats with cerebral ischemia began to drown [14,43]. Additionally, the duration of exercise is used to determine intensity (short duration for low intensity and longer duration for high intensity) increasing confusion between training protocols [14,43]. For instance, it was shown that early submaximal (10 min) swimming is more effective than low (5 min) or maximal “duration-intensity” (20 min) to reduce the escape latency during Morris water maze (MWM) and to increase both vascular endothelial growth factor (VEGF), brain-derived neurotrophic factor (BDNF) levels and antioxidant activity (superoxide dismutase) [14,43]. It has been recently suggested that rodent training protocols should include a physiological indicator of exercise intensity to be extrapolated to humans [44,45]. Yet, empirical running speeds are still frequently used in rodent models, despite the fact that it cannot be applied to clinical studies [14,48,49]. Empirical intensities lead us to consider running speed between 10 and 13 m/min as intense for rats with cerebral ischemia [50], while others postulated that 8 m/min can be considered as slow-to-moderate treadmill training and ~20 m/min as high intensity [51,52].

However, when exercise physiological parameters are used to separate the low- and high-intensity running, moderate running speeds are observed around 17 m/min and the high intensities around 25 m/min in rats with cerebral ischemia [38]. The use of maximal parameters such as VO_2peak_ or maximal speed is practical and can reveal the safe upper limit of subjects. However, stroke-induced physical limitations hamper the capacity of reaching maximal aerobic capacities [2]. To overcome these limitations, a submaximal physiological parameter, the speed associated with the lactate threshold (S_LT_), has recently been used because most individuals with stroke, and all rodents with cerebral ischemia can reach it during an incremental exercise test [2,39,40]. Furthermore, S_LT_ is known to accurately distinguish between high and moderate running speeds, i.e., below S_LT_ for low intensity and above for high intensity [48,49,53]. Considering that, as subjects did not display similar level of aerobic capacity most of the time, intensity needed to be individualized to ensure suitable intensity area.

When comparing various endurance regimens, the session workload is rarely individualized and standardized (work-matched exercise regimens) in both clinical and rodent studies, while it is strongly preconized in stroke exercise guidelines [54,55]. All these parameters are of primary importance to compare exercise doses both to increase the translational relevance of rodent studies and improve the physical exercise recommendations in regard to the time course of brain changes [43,51,56].

### 2.2. Timing of Endurance Training after Stroke

Numerous studies advocated that endurance exercise should be initiated during the acute and subacute phases (first weeks or months) to promote a more effective long-term functional recovery [26,56,57]. Risks of arrhythmia or intracerebral hemorrhage, myocardial injury, systolic dysfunction, unstable angina and uncontrolled hypertension might limit the use of HIIT during the first months (between 1–6 months) in individuals with stroke [12,58]. Nevertheless, individuals with stroke should achieve an incremental exercise test (on treadmill or cycle ergometer) with electrocardiogram monitoring before starting to ensure their safety during training [26,32,33]. Two studies have shown improvements in walking speed when HIIT is performed within the first month [59] or between 3–9 months after stroke onset [32], without measuring its effects on cognitive performance during this period. Despite these encouraging observations, safety of HIIT needed to be confirmed by larger randomized trials involving a wider stroke population panel especially for acute and subacute stroke patients.

When HIIT starts from 6 to >24 months after stroke, no cognitive changes are found, although aerobic capacity and neuroplasticity are improved [35]. In rodents with cerebral ischemia, early HIIT (from day 2 after cerebral ischemia) promotes neuroplasticity, improves functional recovery and reduces depression, thereby suggesting that the acute and subacute phases are suitable in rodents [37,38].

In contrast, a high dose of mobilization (mainly out-of-bed activity) very soon after stroke onset (<24 h) negatively impacts recovery as reported by a large controlled randomized trial [60], which is reinforced by rodent studies [56,57,61]. Accordingly, it is unlikely to be relevant to start HIIT program during this very acute period. Given that both very early or late interventions (<24 h or >24 months after stroke) might mitigate cognition recovery, it is postulated that initiating HIIT within the first months poststroke, i.e., the subacute phase might be more suitable to enhance both sensorimotor and cognitive functions, provided that HIIT should be safe and feasible for individuals with stroke at this period. It also seems important to indicate that the frequency (number of sessions per week) of such exercise types remains understudied.

### 2.3. Blood Measurement of Neurotrophins after Training

In addition, physiological measurements related to neuroplasticity and cognition are also limited in exercise studies. Indeed, for ethical and technical reasons, the most common source for sampling BDNF in humans is peripheral blood [62]. However, circulating BDNF is mainly stored in platelets [63] and comes from many sources such as endothelial cells [64,65], monocytes, B cells, T cells [66] and/or brain [67,68]. BDNF levels in the brain may not be reflected by the amount of BDNF associated with platelets. Hence, it is not surprising that circulating BDNF levels do not mirror brain levels in healthy rats [69]. It explains why interpretation of peripheral BDNF levels is challenging, although the brain is a major origin of the circulating BDNF (70–80% of circulating BDNF) [68]. It, thus, seems that serum BDNF measurement should be combined with complementary measurements (behavioral assessment and/or brain imaging) to better understand the meaning of serum neurotrophic factor levels in exercise protocols. For instance, previous authors have reported that the increased hippocampal volume is correlated with greater serum levels of BDNF and cognitive performance in older individuals [70]. However, cognitive tests are not systematically combined with neuroplasticity measurements in studies on HIIT, thereby limiting evidence on the role of neuroplasticity processes in cognitive improvements after training [21,71,72,73].

Interestingly, a preclinical study has proposed a new method to quantify in vivo the brain BDNF in freely moving mice by collecting microdialysate of cerebrospinal fluid during behavioral tasks or stress condition [74]. This strategy might be used in a specific brain target area throughout a HIIT program in rodents, allowing us to follow the BDNF kinetic on the same animal. However, potential clinical application of such technology in exercise condition still remains difficult to imagine in the short term, although cerebral microdialysis are already performed for monitoring biochemical changes during neurointensive care in humans [75].

## 3. How Can HIIT Promote Neuroplasticity and Cognitive Benefits in Individuals with Stroke?

Several potential molecular factors might mediate the effects of HIIT on neuroplasticity processes and/or cognitive improvements. First, skeletal muscles are able to communicate with other organs such as the brain through many released substances during exercise [76]. Among other substances, lactate is released by active muscles during a HIIT session, which is achieved by healthy people and individuals with stroke [30,77]. An increase in blood lactate concentrations is frequently correlated with upregulation in serum BDNF levels, motor cortex excitability and motor learning in healthy humans [78,79,80]. It is found in mice that lactate originating from active muscles could enter into neurons through its receptor (MCT2) to stimulate BDNF by promoting SIRT1 pathway [81]. The upregulation of hippocampal and cortical BDNF expression and its high-affinity receptor tropomyosin receptor kinase B (TrkB) are well known to promote neurogenesis, neuronal survival and synaptic plasticity and to induce long-term potentiation (LTP) [67,82]. An increase in hippocampal LTP is frequently associated with memory improvements [83]. In the same way, higher BDNF and/or VEGF (neurogenesis and angiogenesis) expression could improve memory performances in healthy rodents after repeated lactate injections to mimic high-intensity exercises [81,84]. Moreover, a blockade of the MCT expression in in vitro experiments reduces the transfer of lactate to astrocytes and neurons and impairs long-term memory in rats [85]. Additionally, lactate infusion at rest could increase circulating BDNF in humans [86].

Moreover, recent evidence suggests a potential role in neuroplasticity after HIIT in both humans and rodents of the endurance exercise-induced myokine, the fibronectin type III domain-containing 5 (FNDC5) [45,87,88,89]. Indeed, Bostrom et al. [88] have observed an upregulation of *Fndc5* gene expression in skeletal muscle and an increase in serum of its secreted form, irisin, after prolonged endurance exercise in mice and humans. It is postulated that irisin itself might be able to cross the blood–brain barrier (BBB) to induce these gene expression changes, or irisin might induce a factor x that can. When hippocampal *Fndc5* is upregulated during training, *Bdnf* and other neuroprotective genes are also activated in the mice hippocampus [89]. Exercise-induced adult hippocampal neurogenesis was associated with increases in both *Fndc5* and *Bdnf* genes, thus improving cognition in a mouse model of Alzheimer’s disease [90].

Then, stroke is associated with a strong neuroinflammation that affects neuroplasticity processes within the core of lesion, the penumbra and the remote areas such as the spinal cord [91,92,93,94]. HIIT might be suitable to reduce pro-inflammatory cytokines in parallel with an activation of microglia (M2 phenotype) in rodents with cerebral ischemia as well as the neurotrophil-to-lymphocyte ratio in patients with multiple sclerosis [38,95].

Finally, it is found that genes associated with the inhibitory neurotransmitter gamma-aminobutyric acid (GABA), which regulate the subgranular zone (SGZ) niche of the stem cells by maintaining their quiescent state, were downregulated in rodents exposed to a 28-day running wheel [96]. The decline in GABA function might elevate BDNF levels that mediates neurogenesis during exercise [97,98,99]. In line with previous results, transcranial magnetic stimulation (TMS) in human studies indicates a decrease in synaptic GABA functions in parallel with improvements in motor memory consolidation after HIIT [100]. In addition, the training-induced synaptic plasticity (GABA) can be investigated through the expression of the potassium–chloride cotransporter (KCC2, a neuronal chloride extruder) and sodium–potassium–chloride cotransporter type 1 (NKCC1, a ubiquitously chloride importer) that are disturbed after cerebral ischemia and lead to alteration in the excitation/inhibition balance in brain [101,102,103]. Evidence has suggested that exercise or mechanical stimulation can alleviate spasticity and neuropathic pain in animal models, likely due to the upregulation of KCC2 expression via the BDNF-TrkB pathway [104]. The molecular processes by which exercise and/or environmental enrichment increase KCC2 levels are still unknown, but endurance training is recognized to upregulate BDNF expression, which is a major determinant of KCC2 upregulation [105]. Similarly, an upregulation of insulin-like growth factor-1 (IGF-1) could decrease the ratio between the expression of NKCC1 and KCC2, promoting the developmental switch of GABA polarity from excitation to inhibition [106]. However, very few studies have assessed the direct effect of HIIT on KCC2 expression after stroke [38].

## 4. Do HIIT Promote Neuroplasticity and Cognitive Benefits in Healthy Individuals and Rodents? Comparison with MICT

Overall, the impact of endurance training on cognitive benefits and cerebral plasticity was often investigated using MICT in healthy individuals [107]. However, discrepancies remain between studies regarding its effectiveness, and the suitable dose (frequency, duration, intensity) of aerobic training is still subject to debate [107,108,109]. In this context, HIIT is frequently compared to MICT to highlight their respective impact on neuroplasticity and/or cognition, but divergent findings remain.

### 4.1. In Humans

In healthy children, a 4-week HIIT program enhances working memory, as observed with the digit span forward and the Tower of Hanoi test performance without modifying other cognitive tasks [110]. Otherwise, most studies have used a single session of HIIT protocols to detect the respective effects of HIIT on neuroplasticity and cognition. Indeed, Winter and colleagues [111] found that a single bout of HIIT speeds vocabulary learning up by 20% contrary to a moderate intensity exercise in healthy sport students. Moreover, serum BDNF, dopamine and epinephrine seem to be important mediators by which HIIT is able to improve retention of the novel vocabulary in this study. Interestingly, when healthy individuals perform two distinct 30-minute sessions (20% below the ventilatory threshold or VT and at 10% above VT), serum BDNF concentrations only increase for the exercise performed above VT (with blood lactate accumulation), while cognitive function scores for the Stroop tests are improved after the two exercise regimens [79]. However, these cognitive improvements are observed without being correlated with BDNF changes in disagreement with the Winter et al. study [79,111]. Similarly, after an acute sprint interval exercise, the shortened response times for both the Stroop task and Trail making test are not correlated with the higher serum BDNF concentrations [112].

An acute bout of high-intensity exercise is able to modulate complex motor behavior by improving motor skill acquisition and memorization in parallel with an increase in some biomarker concentrations (VEGF, IGF-1, BDNF and lactate) [113]. For instance, a HIIT session is effective in increasing long-term retention of the motor skill, serum BDNF concentrations (3.4 fold increase) and LTP-like neuroplasticity when performed immediately after achieving a motor task [114,115]. Additionally, a session with higher intensity, performed immediately before or after practicing a motor task, is more effective than a single session of MICT for increasing long-term retention of motor skill at both 1 and 7 days following learning [116].

Other authors found opposite conclusions by showing higher benefits of an acute MICT session on memory performance than HIIT [117]. A meta-analysis by Chang et al. (2012) showed that lower intensities would better improve cognitive performance immediately after an acute exercise completion (until 1 min) [118]. However, the performance of a cognitive task could be higher than MICT when this task is performed between 11 to 20 min following a single bout of high-intensity exercise. A longer delay would blur positive outcomes on cognitive performance [118]. It, thus, remains difficult to define which type of exercise is the most suitable for cognitive functions. Figure 1 illustrates the effects of HIIT on neuroplasticity processes and cognitive functions in healthy individuals.

### 4.2. In Rodents

On the one hand, an upregulation of TrkB, VEGF and peroxisome proliferator activator receptor γ coactivator-1α (PGC-1α) levels has been recently found following an 8-week HIIT in the rat hippocampus contrary to work-matched MICT. A positive correlation is also observed between the upregulation of *triceps brachii* FNDC5 (fast-twitch muscle fibers) and hippocampal TrkB after HIIT, but not after MICT, in accordance with other authors who have demonstrated a link between myokines and neurotrophins expression [89]. However, Constans et al. failed to detect any effect of HIIT on working and spatial memory [45]. In unpublished results, we have also observed an upregulation of FNDC5 levels in cerebral cortex from 15 days of HIIT as well as higher levels of pTrkB and Pan-neurotrophin receptor p75 (p75^NTR^). Similarly, higher levels of hippocampal BDNF and glial cell-line-derived neurotrophic factor (GDNF) expression are promoted by HIIT compared to MICT [119]. In line with these results, an 8-week endurance training above the lactate threshold (without using HIIT) effectively elicits adult hippocampal neurogenesis in mice by showing an increase in the doublecortin (DCX) and PGC-1α protein expression [120]. Additionally, these authors found a reduction in CCL11 levels, a neurogenesis inhibitory marker, at the end of training reinforcing the potential role of HIIT in hippocampal neurogenesis. It has also been demonstrated that three sessions of HIIT induce an increase in cell proliferation in the hippocampus (minichromosome maintenance complex component 2 or MCM2), immature neuron content (doublecortin or DCX), BDNF and mitochondrial content (voltage-dependent anion-selective channel protein 2, VDAC) [121]. Additionally, a 7-week HIIT induces an increase in both cortical and hippocampal VEGF expression associated with a higher density of blood capillaries, but unfortunately, cognitive outcomes have not been measured [84]. Interestingly, cerebral blood flow is well known to influence cognitive functions in both humans and rodents [122].

A single session of HIIT improves antioxidant mechanisms reducing lipid peroxidation in the hippocampus [123]. Similarly, 6 weeks of HIIT enhance superoxide dismutase concomitantly with an enhanced hippocampal BDNF levels and reduce hippocampal oxidative stress by decreasing lipoperoxidation and cytokine content (TNFα, IL-6, IL-1β and IL-10) [124]. However, these authors failed to find a significant effect on the working memory performance, although HIIT improves cerebellar antioxidant capacity, known to be involved in the higher order behaviors [125].

On the other hand, some authors demonstrated higher effectiveness of MICT to stimulate hippocampal BDNF, IGF-1, VEGF as well as the mitochondrial marker, prohibitin, than a more fatiguing endurance training [71]. Interestingly, although both training paradigms could promote neuronal proliferation and migration in the adult dentate gyrus (DG), moderate but not high-intensity exercise enhanced behavioral spatial discrimination. However, exercise intensity should be considered with caution in this study because both moderate and intense running speeds increase the blood lactate concentration, while MICT should not induce it as mentioned above [30]. Moreover, the intense exercise is an incremental exercise on the treadmill (not an HIIT), which is known to have no effect on the plasma BDNF concentration in young healthy men [126]. Nokia et al. have reported a very modest effect of a 6- to 8-week HIIT on adult hippocampal neurogenesis by showing that the highest number of DCX positive hippocampal cells was observed in rats that ran on a wheel (considered as moderate intensity exercise) [127]. It is noteworthy that the increase in DCX expression at the end of training in the Nokia et al. study [1,2] might not reflect the effect of the entire training period, because its expression needs several weeks to be detected in the hippocampus [128].

## 5. HIIT Could Contribute to Neuroplasticity and Cognitive Recovery after Stroke

### 5.1. Clinical Studies

Figure 1 also illustrates the effects of HIIT on neuroplasticity processes and cognitive functions after stroke. A single bout of HIIT just as a 4-week HIIT program increase serum VEGF and IGF1 levels as well as BDNF levels that are correlated with higher blood lactate concentrations compared to MICT, without a concomitant cortisol stress response (known to limit neuroplasticity processes) [24,35,77,129,130]. In the ipsilesional hemisphere, HIIT induces higher deoxyhemoglobin concentrations compared to MICT, reflecting greater improvements in systemic and cerebral O_2_ consumption, but no cognitive recovery is found [35]. Interestingly, when neuroblastic rat cells in culture are treated with the serum from individuals with stroke achieving HIIT, it results in a higher increase of dendritic growth and mitochondria redistribution along these new dendrites [35]. Other authors also found very low effects of long-term “high-intensity training” (not HIIT in this study) on short-term memory without altering working memory and executive functions [23,27]. Indeed, Tang et al., showed no effect on working memory, attention and conflict resolution when exercise intensity progressively increased from 40 to 80% of HR reserve over 3 months of training by using the Verbal Digit Span Test, Color-Word Stroop test and Trail-Making Test Part B [23] (Table 1).

### 5.2. In Rodents with Cerebral Ischemia

Figure 2 illustrates the effects of HIIT on neuroplasticity processes and cognitive functions after cerebral ischemia. HIIT upregulates the ipsilesional BDNF expression and its high-affinity receptor TrkB in both cortex and hippocampus [37,131]. Indeed, some studies observed higher effects of HIIT on the mBDNF/pro-BDNF ratio compared with work-matched MICT in the ipsilesional CA1, CA3 and DG of the hippocampus. This ratio is closely related to a decline in depression, as shown by using the sucrose preference test [37]. In addition, both aerobic regimens reduced pro-BDNF levels in CA3 and DG regions, which preferentially binds with p75^NTR^, triggering proapoptotic and synaptic withdrawal [37,132]. Using the same protocol based on S_LT_, HIIT could decrease neuronal death in the DG by reducing the expression of the TLR4/NF-kB/NLRP3 pathway and the depression [133]. Earlier poststroke HIIT also downregulates pro- and anti-inflammatory cytokine expression and activates microglia in the ipsilesional hemisphere [38].

However, most studies focused on the ipsilesional side, whereas the contralesional cerebral cortex and hippocampus are also strongly involved in recovery, suggesting that the contralesional side should be considered when assessing treatments [134]. Indeed, inhibiting the contralesional hemisphere with lidocaine after a large ischemic lesion would increase motor deficits of the paretic limb [97,135]. Moreover, higher activity of the contralesional cortex contributes to improve motor activity by reinnervating the spinal cord in mice [93]. Pin-Barre et al. found that HIIT could restore the stroke-induced increase in NKCC1/KCC2 ratio in the contralesional hemisphere in contrast with what is observed in the ipsilesional hemisphere [38]. Additionally, an upregulation of neuroplasticity markers such as TrkB, FNDC5, VEGF and p75^NTR^ is observed in both the contralesional cortex and hippocampus after work-matched short- and long-interval HIIT without significant changes in the ipsilesional side and without gains in cognitive functions [38,39]. Strong improvements in grip strength of the affected forelimb and aerobic parameters are only observed when neuroplasticity markers are increased in the contralesional hemisphere. When no training is performed, insufficient contralesional plasticity occurs together with incomplete functional recovery. The latter study highlights that both long and short HIIT regimens might be used depending on the aerobic abilities and exercise preference of each individual with stroke. For instance, for those who are not able to withstand longer intervals, an individualized HIIT with short intervals can be used without reducing the effectiveness of rehabilitation [26]. Interestingly, it has also been recently found in individuals with stroke that HIIT with short and long intervals is of clinical relevance [136] (Table 2). Unfortunately, no study using HIIT focused on cognitive functions in rodents with cerebral ischemia.

## 6. Perspectives

### 6.1. Is the Combination between HIIT and Cognitive Tasks Effective to Improve Cognitive Performance during the Stroke Rehabilitation?

Both intense and moderate endurance exercises seem to have a modest effect on cognitive recovery [20,23,137]. Nevertheless, it is considered that endurance training might act as a powerful neurogenic stimulus potentiating the effectiveness of cognitive tasks on memory [138]. Indeed, greater cognitive improvements and serum neurotrophic factor upregulation have been reported when endurance training was combined with cognitive tasks such as computerized dual-n-back training [135,139]. In humans, very few studies have examined the combined effects of HIIT and cognitive training. It has been reported that individuals with greater fitness improvements following 6 weeks of combined HIIT and memory training (a computerized version of the concentration memory task) exhibit better high-interference memory performances and greater increases in the serum BDNF and IGF-1 compared with HIIT alone [140]. A combination of various HIIT programs, including cognitive exercises, is effective in young adults for facilitating improvements in aerobic/muscular fitness outcomes and executive functions by using the trail making test (TMT) [137]. Additionally, a single bout of HIIT combined with motor practice could increase skill retention, suggesting a potential impact of HIIT to accelerate motor learning in individuals with stroke [141]. Moreover, priming HIIT through transcranial direct current stimulation enriched with a paretic ankle skill acquisition task could reduce poststroke cortical excitability asymmetry, known to be associated with less functional impairments [[142],[143],[144],[145][146]], which is not observed when HIIT is performed alone [141] (Table 3).

No study in rodents has combined HIIT with cognitive training. However, it is already postulated that the increase in survival of newborn cells within the DG induced by a memory training [147] can be completed by the increased newborn cell proliferation induced by physical exercise training [148]. Indeed, in neurogenesis-ablated mice, the combination of environmental enrichment and exercise partially rescues neurogenesis and restores memory [149].

### 6.2. Pre-Conditioning HIIT Might Reduce Poststroke Brain Damage

Two large prospective studies conducted on more than 20,000 men have shown a decreased risk of stroke incidence when training is considered as vigorous [46,47]. For instance, it is reported that there is a 21% lower risk of stroke when such a session is performed once per week [150], although “vigorous exercise” needs to be taken with caution, as mentioned in methodological considerations. Thus, exercise preconditioning decreases the risk of stroke incidence but can HIIT limit poststroke deleterious outcomes by improving brain ischemic tolerance.

There is little information on pre-stroke (or pre-conditioning) HIIT-induced functional and cognitive changes in rodents with cerebral ischemia. Rezaei et al. are the first to show that an 8-week HIIT can protect BBB integrity, decreasing inflammatory cells infiltration, thereby reducing cortical and total cerebral infarction volumes compared to MICT [151]. Moreover, HIIT promotes higher striatal VEGF-R2 levels (main receptor of VEGF-A) and cortical VEGF-A levels than MICT [151]. To reinforce these findings, higher levels of endothelial nitric oxide synthase (eNOS) and 5’ adenosine monophosphate-activated protein kinase (AMPK) in both brain and cerebral vessels are found after HIIT resulting in a rise in cerebral blood flow and improvements in stroke outcomes by using Bederson score and beam walk tests [152].

Preconditioning HIIT also improves neurological score in rodents by preventing motor deficits and enhancing their recovery [152,153]. This reduction in poststroke deficits might be associated with the rise in BDNF expression in both plasma and brain hemispheres through PGC1α/ERRα (estrogen receptor-related receptor alpha) pathway, known to be involved in both mitochondrial biogenesis and neurotrophin expression [89,153]. In line with these studies, pre-conditioning high-intensity exercise (not HIIT in this study) also reduces infarct edema size at days 1 and 3 poststroke and enhances neuroprotection by decreasing neuronal apoptosis through Heat shock protein 70 (HSP70)/extracellular signal-regulated kinases 1 and 2 (ERK1/2) cascade, when the lesion occurs within the 24 h after the last training session [154]. Interestingly, different outcomes were found according to the delay between the last HIIT session and stroke onset. The more the delay increases, the more the infarct and edema sizes increase with worsened functional outcomes [152]. The preventive role of HIIT, thus, seems to be a promising research area in the context of stroke.

### 6.3. Is the HIIT Effectiveness on Neuroplasticity/Cognition Observed in Other Neurologic Disorders?

Recent studies have reported promising effects of HIIT on cognitive functions in neurodegenerative diseases. In patients with middle cognitive impairments, HIIT combined to a ketogenic diet and memory training may reverse early stage memory loss [155]. Moreover, HIIT decreases depression in people with severe mental illness [156]. Depressive symptoms are reduced after an 8-week HIIT in healthy women [157]. Furthermore, HIIT is effective in alleviating cognitive decline in Alzheimer’s disease mice through improvements in hippocampal mitochondrial morphology together with the reduction in mitochondrial fragmentation and hippocampal β-Amyloid burden [158]. In line with previous results, HIIT protects rats from post-traumatic stress disorder memory decline by decreasing oxidative stress, anxiety levels and by improving antioxidant capacity, therefore reducing neuronal damage [159]. In patients with multiple sclerosis, HIIT reduces inflammation and enhances in parallel executive functions as well as verbal memory. Nevertheless, no significant cognitive nor quality-of-life improvements are observed in people with Parkinson’s disease after HIIT, while it is effective in improving serum BDNF and other functional outcomes [160,161,162].

## 7. Conclusions

It seems that HIIT should be included in stroke rehabilitation for its beneficial effects on neuroplasticity processes. The clinical role of neuroplasticity observed in each hemisphere needs to be clarified by coupling more frequently cellular/molecular measurements and behavioral testing. Despite these results, HIIT induces very modest cognitive effects when performed alone in both healthy people and individuals with stroke. However, its powerful neurogenic effect might help to accentuate benefits induced by cognitive tasks. Based on these considerations, it is recommended to continue investigating the different modalities of HIIT on brain plasticity in terms of duration and/or intensity of both high-intensity intervals and recovery phases along with the type of recovery between series (active or passive) and the mode of HIIT exercise (cycling, running, swimming, rowing, etc.). It is noteworthy that this review is not designed to demonstrate a useless/ineffective role of MICT after stroke. In contrast, we believe that both low- and high-intensity training regimens might be complementary for brain health.

## Figures and Tables

**Figure 1 ijms-22-03003-f001:**
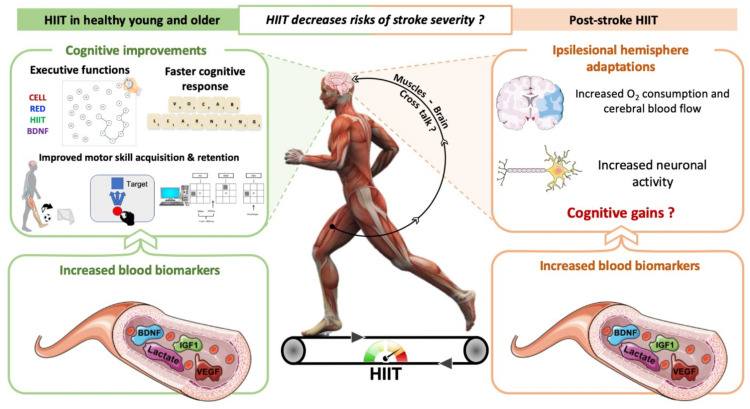
Overview of the influence of high-intensity interval training (HIIT) on neuroplasticity and learning/memory performance in healthy humans and individuals with stroke. HIIT enhances circulating biomarker expression of neuroplasticity processes in individuals with stroke. The HIIT effects on cognitive functions remain to be defined despite some authors finding benefits in cognitive performance in healthy individuals.

**Figure 2 ijms-22-03003-f002:**
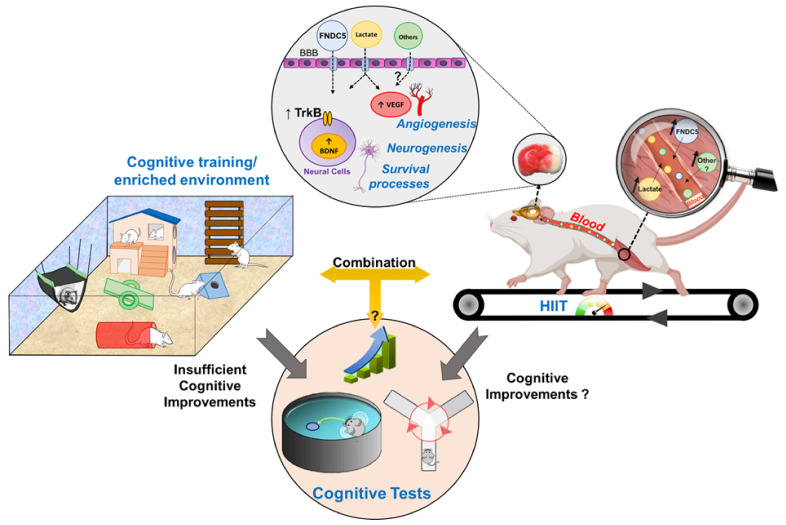
Overview of the influence of high-intensity interval training (HIIT) and cognitive training on neuroplasticity and learning/memory performance in rodents with cerebral ischemia. HIIT enhances neurotrophin expression, neurogenesis and synaptic plasticity. However, the effects on cognition remain unclear but seem to be very modest. It is unknown if the combination between HIIT and cognitive training (enriched environment) can increase benefits of an enriched environment on cognitive functions. Grey arrows mean effects of each type of training on cognitive functions (in the beige circle at the bottom). “?” means that it remains unknown (no evidence).

**Table 1 ijms-22-03003-t001:** Summary of aerobic training protocols and their effects on cognition in stroke patient.

Studies	Participants	Aerobic Training		Results
		Intensity	Duration	
Tang et al., 2016 [23]	**n** = 25 **Age**: 66 (62–71) years **Timing after stroke:** 3.5 (2.2–6.7) years	From 40 to 80% HRR	60 min/session 3 sessions/week for 6 months	➚ Short-term memory ⟺ Working memory, set shifting, conflict resolution
Boyne et al., 2019 [77]	**n** = 16 **Age:** 57.4 (37.7–72.1) years **Timing after stroke:** 6.5 (0.5–16.11) years	Treadmill HIIT: maximum tolerated speed 30 sec HI and 60 to 30 s LI Seated Stepper HIIT: maximal cadence with 50% of maximal resistance MICT: 45 ± 5% HRR	25 min/session	➚ serum BDNF Lower ➚ in serum BDNF after MICT
Boyne et al., 2020 [24]	**n** = 16 **Age:** 57.4 (37.7–72.1) years **Timing after stroke:** 6.5 (0.5–16.11) years	Treadmill HIIT: maximum tolerated speed 30 sec HI and 60 to 30 s LI Seated Stepper HIIT: maximal cadence with 50% of maximal resistance MICT: 45 ± 5% HRR	25 min/session	➚ VEGF, IGF-1 after HIIT ➚ Serum BDNF is correlated to ➚ blood lactate after HIIT
Hsu et al., 2020 [35]	**n** = 28 **Age:** HIIT: 58.5 (49.8–67.2) years MICT: 53.1 (46.2–60.0) years **Timing after stroke:** 38.5 (19.1–57.9) months	Bicycle ergometer HIIT: 3 min at 80% VO_2peak_ separated by 3 min at 40% VO_2peak_ Bicycle ergometer MICT: 60% VO_2peak_	30 min/session Isocaloric 2 to 3/week 36 sessions	➚ VO_2peak_ after HIIT > MICT ➚ peak cardiac output ➚ △[HHB] and △[THB] after HIIT in lesioned hemisphere ➚ Serum BDNF after HIIT ➚ Dendritic growth with patient serum after HIIT

➚ indicate an increase; ⟺ indicate a maintenance; BDNF: brain-derived neurotrophic factor; VEGF: vascular endothelial growth factor; IGF-1: insulin-like growth factor 1; HRR: heart rate reserve; HI: high-intensity; LI; low-intensity; HIIT: high-intensity interval training; MICT: moderate-intensity.

**Table 2 ijms-22-03003-t002:** Summary of aerobic training protocols and their effects on cognition in poststroke rats.

Studies	Participants	Aerobic Training		Results
		Intensity	Duration	
Pin-Barre et al., 2017 [38]	Sprague-Dawley n = 70 **Age:** 2–3 months **Method:** tMCAO (120 min) **Timing after stroke:** 24–48 h	HIIT: 4 × (4 + 3 min active rest) 80% of S_max_-S_LT_ (week 1) 95% of S_max_-S_LT_ (2) MICT: 80% S_LT_	28 min/session Isocaloric 5/week for 2 weeks	➚ Endurance performance after HIIT ➘ Inflammation mainly in the lesioned hemisphere Restored NKCC1/KCC2 ratio in the contralesional hemisphere
Luo et al., 2018 [37]	Wistar n = 55 **Age:** 2–3 months **Method:** tMCAO (90 min) Timing after stroke: 28 days	HIIT: 4 × (4 + 3 min rest) S_LT_ + 60–70% (S_max_-S_LT_) MICT: 80–90% S_LT_	28 min/session Isocaloric 5/week for 4 weeks	➚ BDNF in ipsilesional CA1, CA3 and DG after HIIT ➚ mBDNF/proBDNF ratio in hippocampus after HIIT ➚ TrkB and NR2A expression after HIIT ➘ p75^NTR^ and NR2B after HIIT
Li et al., 2020 [133]	C57BL/6J mice n = 5/group **Age:** 8–10 weeks **Method:** tMCAO (90 min) **Timing after stroke:** 28 days	HIIT: 4 × 4 (4 + 3 min rest) S_LT_ + 60–70% (S_max_-S_LT_) MICT: 80% S_LT_	28 min/session Isocaloric HIIT: 5/week MICT: 7/week for 4 weeks	➘ Neuronal death in DG after HIIT ➚ Neuroprotection through ➘ PTEN activity after HIIT ➘ Depression-like behavior after HIIT
Pin-Barre et al., 2021 [39]	Sprague-Dawley n = 42 **Age:** 2–3 months **Method:** tMCAO (120 min) **Timing after stroke:** 24–48 h	HIIT4: 4 x (4 + 3 min active rest) HIIT1: 1 + 1 min active rest 80% of S_max_ – S_LT_ (1st week) 95% of S_max_ – S_LT_ (2nd week)	28 min/session Isocaloric 5/week for 2 weeks	Both HIIT does not reduce stroke-induced gliogenesis in the ipsilesional hesmisphere Both HIIT ➚ pTrkB in the contralesional hippocampus while HIIT4 only ➚ pTrkB in the contralesional cortex Both HIIT ➚ FNDC5 and Cyt C in the contralesional cortex

➚ indicate an increase; ➘ indicate a decrease; BDNF: brain-derived neurotrophic factor; proBDNF: precursor brain-derived neurotrophic factor; mBDNF: mature brain-derived neurotrophic factor; VEGF: vascular endothelial growth factor; IGF-1: insulin-like growth factor 1; NKCC1: Na^+^–K^+^–2Cl^−^ cotransporter; KCC2: K^+^–Cl^−^ cotransporter; HRR: heart rate reserve; HI: high-intensity; LI; low-intensity; HIIT: high-intensity interval training; MICT: moderate-intensity continue training; HHb: deoxyhemoglobin; THb: total hemoglobin; S_LT_: speed at lactate threshold; S_max_: maximal speed; tMCAO: transient middle cerebral artery occlusion; DG: dentate gyrus; TrkB: *Tropomyosin receptor kinase B*; p75^NTR^: p75 neurotrophin *receptor; NR2A:* N-methyl-D-aspartate subtype glutamate receptor leading to LTP; *NR2B:* N-methyl-D-aspartate subtype glutamate receptor producing LTD, pTrkB: phosphorylated form of tropomyosin receptor kinase B; FNDC5: fibronectin type III domain-containing protein 5; Cyt C: Cytochrome C.

**Table 3 ijms-22-03003-t003:** Summary of HIIT/intense protocols combination and their effects on cognition in healthy people.

Studies	Participants	Aerobic Training		Combination	Results
		Intensity	Duration		
Madhavan et al., 2016 [144]	n = 11 **Age:** 58 **Timing after stroke:** 9 years	Incremental walking speed until 80% of the age-predicted HR (220-age)	40 min/session 1 session	tDCS enhanced with a paretic ankle skill acquisition task (15 min)	➘ CME of the paretic tibialis anterior after HIIT alone ➚ CME of the paretic tibialis anterior after the combination ➘ RPE after combination
Nepveu et al., 2017 [141]	n = 22 **Age:** 64.9 **Timing after stroke:** chronic stroke patients	HIIT: 3 × 3 min at 100% peak workload GXT interspersed with 2 × 2 min at 25%	15 min/session 1 session	Time-on-target motor task ending 10 min before HIIT initiation Retention test 24 h after HIIT session	➘ Tendency of SICI measured by TMS ➚ Skill retention after HIIT
Madhavan et al., 2020 [143]	n = 81 **Age:** 58.8 **Timing after stroke:** 5.5 years	Speed increment over 2 min to reach the maximal speed for 10 s Warm-up HR during recovery initiate a new interval	40 min/day 3 days/week for 4 weeks	tDCS enhanced with a paretic ankle skill acquisition task (15 min)	CME with the combination Patients with ➚ CME increased walking speed more than others

➚ indicate an increase; ➘ indicate a decrease; HR: heart rate; HIIT: high-intensity interval training; RPE: rate of perceived exertion, GXT: graded exercise test; tDCS: transcranial direct current stimulation; TMS: transcranial magnetic stimulation; CME: corticomotor excitability; SICI: short-interval intracortical inhibition.

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
