# Peer review of "Is High-Intensity Interval Training Suitable to Promote Neuroplasticity and Cognitive Functions after Stroke?"

_ijms, 2021, doi:10.3390/ijms22063003_

Round 1
Reviewer 1 Report
Dear Authors
The authors presented a thorough explanation of the effects of the HITT and addressed, the gap between present animal studies on HITT and on the limited and heterogeneous results presented from clinical studies.
One point that needs to be addressed is the discrepant findings presented by the final AVERT trial ( Lancet. 2015 ) where the clinical trial data showed no strong major benefits of early and higher dose mobilization within 24 hours of stroke compared to the control group.
Because the AVERT trial is one of the representative studies of providing early exercise in stroke patients, this reference needs to be cited and some postulated theories from the authors' point of view for these differences would be insightful.
Author Response
#Reviewer 1
The authors presented a thorough explanation of the effects of the HITT and addressed, the gap between present animal studies on HITT and on the limited and heterogeneous results presented from clinical studies.
Response to suggestion of the reviewer.
- One point that needs to be addressed is the discrepant findings presented by the final AVERT trial (Lancet. 2015) where the clinical trial data showed no strong major benefits of early and higher dose mobilization within 24 hours of stroke compared to the control group.
Because the AVERT trial is one of the representative studies of providing early exercise in stroke patients, this reference needs to be cited and some postulated theories from the authors' point of view for these differences would be insightful.
In accordance with the reviewer, we added some information about effects of very early mobilization after cerebral ischemia in both stroke patients and rodent models although it is not directly related to endurance training. We indicated that it is not relevant at this stage.
Paragraph “2.2. Timing of endurance training after stroke”.
Line 158: “In contrast, high dose of mobilization (mainly out-of-bed activity) in very early time after stroke onset (< 24 h) negatively impacts recovery as reported by a large controlled randomized trial [1], which is reinforced by rodent studies [2–4]. Accordingly, it is unlikely to be relevant to start HIIT program during this very acute period. Given that both very early or late interventions (<24 h or >24 months after stroke) might mitigate cognition recovery, it is postulated that initiating HIIT within the first months poststroke, i.e., the sub-acute phase might be more suitable to enhance both sensorimotor and cognitive functions, provided that HIIT should be safe and feasible for individuals with stroke at this period. It seems also important to indicate that the frequency (number of sessions per week) of such exercise type remains understudied.”
Reviewer 2 Report
Present evidence suggests that HIIT is a safe approach to trigger neurorehabilitation although this has yet to be shown to be absolutely safe in subjects with subacute and chronic stroke and those with and without other comorbidities. Therefore, this topic has to be thoroughly investigated in high quality randomized controlled studies before any particular conclusions are drawn out. This point has to be also included in the discussion.
Preliminary results from randomized trials suggest that HIIT may be effective in improving mobility and gait outcomes, cardiovascular health, angiogenesis with evidence of neuroplasticity especially in the primary motor cortex.
To date, evidence surrounding whether cardiovascular exercise can have positive effects on cognition shows mixed results. Cardiovascular exercise has been shown to improve cognition in healthy individuals (Hillman et al., 2008) through long term structural and functional synaptic changes including hippocampal neurogenesis (Thomas et al., 2012). However, a recent Cochrane review suggests no evidence that aerobic physical activities that improve cardiorespiratory fitness, have any cognitive benefit in healthy older adults (Young et al., 2015).
It still remains to be explored whether BDNF can be used as a surrogate marker of CNS plasticity and it being complexed in platelets might not be ideally seen to have a direct role in the brain. Studies have shown that levels of BDNF can be sampled directly from the rodent brain with an interdigitated microelectrode (IME) biosensor (Yoo et al., 2016) and therefore I assume that these same levels could be studied in a post stroke rodent model that is trained with HIIT on an exercise platform.
Author Response
#Reviewer 2
Response to suggestions of the reviewer.
- Present evidence suggests that HIIT is a safe approach to trigger neurorehabilitation although this has yet to be shown to be absolutely safe in subjects with subacute and chronic stroke and those with and without other comorbidities. Therefore, this topic has to be thoroughly investigated in high quality randomized controlled studies before any particular conclusions are drawn out. This point has to be also included in the discussion.
We agree with the reviewer comment. We have qualified our statements concerning the safety of HIIT in stroke population.
Line 150: “Despite these encouraging observations, safety of HIIT needed to be confirmed by larger randomized trials involving wider stroke population panel especially for acute and subacute stroke patients.”
- To date, evidence surrounding whether cardiovascular exercise can have positive effects on cognition shows mixed results. Cardiovascular exercise has been shown to improve cognition in healthy individuals (Hillman et al., 2008) through long term structural and functional synaptic changes including hippocampal neurogenesis (Thomas et al., 2012). However, a recent Cochrane review suggests no evidence that aerobic physical activities that improve cardiorespiratory fitness, have any cognitive benefit in healthy older adults (Young et al., 2015).
We agree with the reviewer. Our statement was focused on specific effects of HIIT on cognitive functions and associated neuroplasticity but we added more information concerning divergence between studies about traditional aerobic training.
Paragraph 4. Do HIIT promote neuroplasticity and cognitive benefits in healthy individuals? Comparison with MICT
Line 246 : “Overall, the impact of endurance training on cognitive benefits and cerebral plasticity was often investigated using MICT in healthy individuals [5]. However, discrepancies remain between studies regarding its effectiveness and the suitable dose (frequency, duration, intensity) of aerobic training is still subject to debate [5–7]. In this context, HIIT…”
- It still remains to be explored whether BDNF can be used as a surrogate marker of CNS plasticity and it being complexed in platelets might not be ideally seen to have a direct role in the brain. Studies have shown that levels of BDNF can be sampled directly from the rodent brain with an interdigitated microelectrode (IME) biosensor (Yoo et al., 2016) and therefore I assume that these same levels could be studied in a post stroke rodent model that is trained with HIIT on an exercise platform.
As we have indicated in the manuscript, the BDNF quantification remains challenging within human brain and needs to be developed. As well, in vivo monitoring of biochemical changes in animal models could help to deepen the impact of various therapies on neuroplasticity and associated behavioral alteration after brain diseases. In this context, Yoo et al, (2016) provided some potential perspectives in the field and it appears interesting to mentioned it. We thus completed our statement according to the reviewer suggestion.
Paragraph “2.3. Blood measurement of neurotrophins after training”
Interestingly, a preclinical study has proposed a new method to quantify in vivo the brain BDNF in freely moving mice by collecting microdialysate of cerebrospinal fluid during behavioral tasks or stress condition [79]. This strategy might be used in a specific brain target area throughout HIIT program in rodents allowing to follow the BDNF kinetic on the same animal. However, potential clinical application of such technology in exercise condition remains still difficult to imagine in the short term although cerebral microdialysis are already performed for monitoring biochemical changes during neurointensive care in Human [80].